# STAT3 and NTRK2 Genes Predicted by the Bioinformatics Approach May Play Important Roles in the Pathogenesis of Multiple Sclerosis and Obsessive–Compulsive Disorder

**DOI:** 10.3390/jpm12071043

**Published:** 2022-06-26

**Authors:** Ali Sepehrinezhad, Ali Shahbazi, Ali Bozorgmehr, Babak Kateb, Vicky Yamamoto, Sajad Sahab Negah

**Affiliations:** 1Department of Neuroscience, Faculty of Advanced Technologies in Medicine, Iran University of Medical Sciences, Tehran 1449614535, Iran; sepehrinezhad.a@iums.ac.ir; 2Neuroscience Research Center, Mashhad University of Medical Sciences, Mashhad 9919991766, Iran; 3Cellular and Molecular Research Center, Iran University of Medical Sciences, Tehran 1449614535, Iran; 4Iran Psychiatric Hospital, Iran University of Medical Sciences, Tehran 1449614535, Iran; bozorgmehr.a@iums.ac.ir; 5Middle East Brain + Initiative, Los Angeles, CA 90272, USA; babak.kateb@worldbrainmapping.org; 6National Center for Nanobioelectronics, Los Angeles, CA 90272, USA; 7Society for Brain Mapping and Therapeutics, Los Angeles, CA 90272, USA; vicky.yamamoto@usc.edu; 8Brain Technology and Innovation Park, Los Angeles, CA 90272, USA; 9USC-Norris Comprehensive Cancer Center, USC-Keck School of Medicine, Los Angeles, CA 90033, USA; 10Department of Neuroscience, Faculty of Medicine, Mashhad University of Medical Sciences, Mashhad 9919991766, Iran; 11Shefa Neuroscience Research Center, Khatam Alanbia Hospital, Tehran 1996835911, Iran

**Keywords:** multiple sclerosis, obsessive–compulsive disorder, computational biology, molecular function, signaling pathway, microRNAs

## Abstract

**Background:** There are no data available on the levels of genetic networks between obsessive–compulsive disorder (OCD) and multiple sclerosis (MS). To this point, we aimed to investigate common mechanisms and pathways using bioinformatics approaches to find novel genes that may be involved in the pathogenesis of OCD in MS. **Methods:** To obtain gene–gene interactions for MS and OCD, the STRING database was used. Cytoscape was then used to reconstruct and visualize graphs. Then, ToppGene and Enrichr were used to identify the main pathological processes and pathways involved in MS-OCD novel genes. Additionally, to predict transcription factors and microRNAs (miRNAs), the Enrichr database and miRDB database were used, respectively. **Results:** Our bioinformatics analysis showed that the signal transducer and the activator of transcription 3 (*STAT3*) and neurotrophic receptor tyrosine kinase 2 (*NTRK2*) genes had connections with 32 shared genes between MS and OCD. Furthermore, *STAT3* and *NTRK2* had the greatest enrichment parameters (i.e., molecular function, cellular components, and signaling pathways) among ten hub genes. **Conclusions:** To summarize, data from our bioinformatics analysis showed that there was a significant overlap in the genetic components of MS and OCD. The findings from this study make two contributions to future studies. First, predicted mechanisms related to *STAT3* and *NTRK2* in the context of MS and OCD can be investigated for pharmacological interventions. Second, predicted miRNAs related to *STAT3* and *NTRK2* can be tested as biomarkers in MS with OCD comorbidity. However, our study involved bioinformatics research; therefore, considerable experimental work (e.g., postmortem studies, case–control studies, and cohort studies) will need to be conducted to determine the etiology of OCD in MS from a mechanistic view.

## 1. Introduction

Multiple sclerosis (MS) is a demyelinating disease characterized by a wide variety of symptoms, involving motor and cognitive systems. Psychiatric problems are common in MS patients and have a significant influence on the progression of the disease, disability, and quality of life. The main psychiatric comorbidities in MS patients are obsessive–compulsive disorders (OCD), specific phobias, depression, generalized anxiety, and schizophrenic and bipolar disorders [1,2]. The frequency of OCD among MS patients has been reported to be about 12%–16% [3,4] or even 30% in some populations such as in Saudi Arabia [5]. A recent descriptive study reported the experience of OCD in 15 patients with MS [6]. OCD is an anxiety disorder that can be disabling and chronic if it remains untreated. OCD is characterized by a combination of consuming obsessions (i.e., intrusive thoughts or images caused by severe distress) and compulsions that are repetitive behaviors for decreasing anxiety [7].

The precise etiology of OCD’s coexistence in MS is not clear, but it has been suggested that the psychiatric comorbidity is the result of distraction of the connection between different brain regions [3]. Moreover, OCD symptoms are deteriorated by structural brain changes which include: reduced gray matter volume in the right inferior and middle temporal gyri and the inferior frontal gyrus; and the appearance of a right parietal white matter MS plaque [8,9]. To date, most of the research aimed at clarifying clinical symptoms between OCD and MS has focused on functional circuits and structural abnormalities [10,11]. However, understanding of the mechanisms underpinning these commonalities is presently inadequate. Genetic factors have been linked to the risk of developing OCD. For example, it has been reported that variants in different genes, such as solute carrier family 6 member 4 (*SLC6A4*) [12], glutamate ionotropic receptor kainate type subunit 2 (*GRIK2*) [13], monoamine oxidase A (MAOA) [14], dopamine receptor D4 (DRD4) [15], catechol-O-methyltransferase (*COMT*) [16], and brain-derived neurotrophic factor (*BDNF*) [17] are correlated with the risk of OCD in different populations. Therefore, further studies at the level of genes and molecular underpinnings are warranted to figure out the common pathogenesis of OCD and MS. Bioinformatics methods based on data from prior knowledge can also be very valuable for biological applications. 

This study reports the first comprehensive bioinformatics analysis of the connections between OCD and MS at the level of the genetic network, biological processes, and molecular functions. To this point, we reconstructed a new network for common genes between MS and OCD by analyzing topological and physical interactions, such as degree, closeness centrality, and betweenness centrality. The most surprising aspect of our study was that *STAT3* and *NTRK2* genes had the highest connections with common genes between MS and OCD. Moreover, *STAT3* and *NTRK2* had the greatest centrality among ten novel genes. They had the maximum enrichment results, such as molecular function and pathways when compared with the other novel predicted genes. 

## 2. Materials and Methods

### 2.1. Study Design 

As a first step, all genes associated with both MS and OCD are listed (Appendix A). Next, shared genes with the highest topological features were extracted. Then, to find out the genes that have significant interaction with 32 shared genes, the STRING database was used. Since these genes have strong connections with common genes, it is suggested that molecular underpinnings, protein–protein interactions, and cellular changes of novel genes can give us new insights into pathological features between MS and OCD. To this point, further steps of analysis were performed on 10 genes that have close connections with shared genes. To identify protein–protein interactions, target genes were uploaded into the STRING database (https://string-db.org/ (accessed on 24 December 2021)). Target genes were uploaded into Cytoscape to predict all functional interactions and to obtain the main network of topological features. To find out pathways in target genes, WikiPathways analysis was used. We also used Enricher to analyze the consequences of target genes on cell type and brain region. Finally, a link between the pathological processes of MS and OCD is recognized at the transcriptional and miRNA levels. These analyses help us to identify similar genetic and biological features between MS and OCD and give some cues to comprehend significant pathological mechanisms (Figure 1). 

### 2.2. Gene Set Selection 

To obtain genes associated with MS and OCD, two batches of literature-based disease–gene relation data and gene data sets (updated in 2021) were integrated. For this purpose, a comprehensive literature review was performed in PubMed as follows: 

(Multiple sclerosis and linkage, MS and linkage, Disseminated and linkage, Multiple sclerosis and genetic, MS and genetic, Disseminated and genetic, Multiple sclerosis and association, MS and association, Disseminated and association, Multiple sclerosis and GWAS, MS and GWAS, Disseminated and GWAS, Multiple sclerosis and genome-wide association, MS and genome-wide association, Disseminated and genome-wide association, Obsessive–Compulsive and linkage, Obsessive–Compulsive disorder and linkage, OCD and linkage, Obsessive–Compulsive and genetic, Obsessive–Compulsive disorder and genetic, OCD and genetic, Obsessive–Compulsive and GWAS, Obsessive–Compulsive disorder and GWAS, OCD and GWAS, Obsessive–Compulsive and genome-wide association, Obsessive–Compulsive disorder and genome-wide association, OCD and genome-wide association). The genes were inserted into two separated tab pages in an Excel file. We also extracted all genes from two important datasets in the Harmonizome database (https://maayanlab.cloud/Harmonizome/ (accessed on 15 November 2021)) as Gene–Disease Associations (GAD) and Gene–Disease Associations (CTD) for MS and OCD, respectively. These genes were also uploaded into two separated tab pages of an Excel file. In the next step, overlap genes between the literature review and Harmonizome database were removed. Finally, common genes between MS-associated and OCD-associated genes were identified and saved for further analysis. Common genes were then submitted to the STRING database after selecting Homo sapiens organism and 0.400 medium confidence. Then, unconnected genes were excluded, and the top-ten genes were predicted for shared genes based on co-expression, text mining, experiments, databases, gene fusion, co-occurrence, and protein–protein interactions through the STRING database.

### 2.3. Genetic Network Reconstruction Using Cytoscape 

In the current study, the STRING database was used to construct networks and visualize different interactions [18,19]. Next, to investigate the main possible genetic connections and interactions, networks were uploaded into Cytoscape [20,21]. Afterward, the Network Analyzer Toolkit was used to visualize gene connections with nodes and edges. Finally, some basic parameters (i.e., the number of nodes and edges) and topological features (i.e., diameter, density, and centralization) were estimated for each gene set, especially the novel gene set. Centrality parameters were used to show the interactions of the genes in each network (Supplementary BOX S1). 

### 2.4. TRANSFAC Analysis and microRNA Target Prediction 

We used Enrichr (https://maayanlab.cloud/Enrichr/ (accessed on 27 December 2021)) for predicting some significant transcription factors via TRANSFAC and the JASPAR PWMs panel about MS and OCD novel genes. Importantly, for miRNA target prediction, we inserted MS–OCD-associated novel genes into the miRDB database (http://mirdb.org/mining.html (accessed on 27 December 2021)). miRDB is a database for predicting functional miRNAs and annotations of gene targets [22]. We only considered miRNAs with a target prediction score of greater than 90% with human species.

### 2.5. Gene Ontology Enrichment Analysis

Gene set enrichment analysis (GSEA) is used for statistical analysis of gene groups that are over-represented in a large set of genes and may be involved in the pathogenesis of many disorders and disease phenotyping. [23]. Enrichr, Gene Ontology (GO) Consortium (http://www.geneontology.org/ (accessed on 29 December 2021)) and ToppGene databases (https://toppgene.cchmc.org/ (accessed on 29 December 2021)) were applied to perform gene set enrichment analysis. Afterward, some ontologies such as biological processes and molecular functions, related to the individual gene set were statistically analyzed [24,25]. To identify the most important pathways involved, our target genes were uploaded to the WikiPathways database. WikiPathways is a platform and database for creating and enriching biological pathway diagrams for input genes [26].

## 3. Results

### 3.1. Finding Genes According to the Literature Review and Harmonizome 

By searching the available articles and extracting data from the Harmonizome database, we prepared 660 genes that were associated with MS and 191 genes concerning OCD (Table 1; detailed information is in Appendix A). Among them, 32 genes were common between MS and OCD (Figure 2A). We also predicted 10 top genes that had strong connections with 32 common genes between MS and OCD (Figure 2B). Further steps of analysis were performed on 10 top genes that have close connections with shared genes.

### 3.2. Genetic Network Reconstruction

Amongst the genes related to MS and OCD, there was no connection for 48 and 19 genes, respectively. Based on a topological feature, some important genes, such as tumor protein p53 (*TP53*), interleukin 6 (*IL-6*), tumor necrosis factor (*TNF*), epidermal growth factor receptor (*EGFR*), mitogen-activated protein kinase 1 (*MAPK1*), brain-derived neurotrophic factor (*BDNF*), c-myc (*MYC*), and interleukin 2 (*IL-2*) were the most determinant nodes in the MS gene set (Appendix A). On the other hand, the OCD network is comprised of 172 interacted nodes (Appendix A), among them, *BDNF*, serine-threonine protein kinase *AKT1*, dopamine receptor D2 (*DRD2*), and *IL-6* had the largest betweenness centrality. The MS-OCD-associated genetic network had 32 nodes and 35 edges (Figure 2A). Glypican-6 (*GPC6*) had no connection with other nodes in the network. As shown in Figure 2A, the highest degree and maximum closeness and betweenness centrality were detected for 10 shared genes. Among them, BDNF as a neurotropic growth factor (degree (D): 30; betweenness centrality (B): 0.35058201; closeness centrality (C): 0.78431373), IL-6 as an inflammatory cytokine (D: 22; B: 0.08110394; C: 0.66666667), caspase 3 as an apoptotic marker (D: 21; B: 0.07726389; C: 0.64516129), TNF as an inflammatory cytokine (D: 20; B: 0.05743113; C: 0.64516129), MAPK1 as an extracellular signal-regulated kinase (D: 18; B: 0.02775084; C: 0.61538462), estrogen receptor 1 (ESR1) (D: 15; B: 0.06606276; C: 0.58823529), tyrosine hydroxylase as a catalyzing enzyme (TH) (D: 15; B: 0.02649319; C: 0.58823529), nuclear receptor subfamily 3 group C member 1 (NR3C1) as glucocorticoid receptor (D: 15; B: 0.01322346; C: 0.58823529), cannabinoid receptor 1 (CNR1) (D: 13; B: 0.01828817; C: 0.56338028), and catenin beta 1 (CTNNB1) (D: 9; B: 0.01996046; C: 0.54054054) had the highest topological features (Figure 2A). According to available protein–protein interaction data in the STRING database, we predicted 10 top genes, including interleukin 10 (*IL-10*), Signal transducer and activator of transcription 3 (*STAT3*), interleukin 4 (*IL-4*), neurotrophic receptor tyrosine kinase 2 (*NTRK2*), neuroligin-1 (*NLGN1*), neuroligin 4 X-Linked (*NLGN4X*), axin 1(*AXIN1*), TNF receptor superfamily member 1A (*TNFRSF1A*), ephrin type-A receptor 1 (*EPHA1*), and neuroligin 3 (*NLGN3*) as the most related involved novel genes in MS and OCD (Figure 2B).

### 3.3. Predicted Transcription Factors and miRNAs for Hub Genes 

Our human transcription factor analysis predicted five significant transcription factors including nuclear transcription factor Y subunit alpha (NFYA) (predicted by *NTRK2*, *NLGN1*, *NLGN4X*, *AXIN1*, and *TFNRSF1A* genes), core-binding factor subunit beta (CBFB) (predicted by *STAT3*, *NTRK2*, and *EPHA1* genes), transcription factor AP-2 gamma (TFAP2C) (predicted by *AXIN1*, *EPHA1*, and *NLGN3* genes), zinc finger protein 148 (ZNF148) (predicted by *NLGN4X*, *AXIN1*, and *TFNRSF1A* genes), and nuclear factor 1 C-type (NFIC) (predicted by *STAT3*, *NLGN1*, *NLGN4X*, *TFNRSF1A*, and *EPHA1* genes) for target genes (Figure 3A). Importantly, we predicted 26 significant miRNAs with a target prediction score of greater than 90 percent for MS-OCD-associated novel genes (Figure 3B). 

### 3.4. Gene Ontology Enrichment Analysis

Biological process enrichment analysis indicated that presynaptic membrane assembly, postsynaptic membrane assembly, regulation of chronic inflammatory response, neuron cell–cell adhesion, positive regulation of developmental process, regulation of multicellular organismal development, regulation of anatomical structure morphogenesis, and positive regulation of synaptic transmission, glutamatergic could be considered as the disrupted key processes in MS and OCD (Table 2).

As shown in Figure 4, enrichment parameters including molecular function and pathways were predicted for ten hub genes that had the highest connections with common genes. Among them, *STAT3* and *NTRK2* had the maximum enrichment parameters in terms of molecular function and pathways. In molecular function variables, protein homodimerization activity was predicted for *STAT3* and *NTRK2*. However, protein kinase binding and primary miRNA binding were predicted for *STAT3*. Transmembrane receptor protein kinase activity and neurotrophin binding were predicted for *NTRK2* in terms of molecular function. 

The main disrupted pathways were cell migration and invasion through p75NTR, mBDNF and proBDNF regulation of GABA neurotransmission, and BDNF signaling pathways were predicted as the common pathways for *STAT3* and *NTRK2*. Furthermore, the major disrupted pathways for *STAT3* were cytokines (i.e., IL-4, 10, 7, 9, and 17), Interferon type I signaling pathways, TGF-beta receptor signaling, and dopaminergic neurogenesis. Moreover, key disrupted pathways for *NTRK2* were MAPK signaling pathway, PI3K-Akt signaling pathway, and BDNF-TrkB signaling (Figure 4). 

## 4. Discussion

MS and OCD are complicated diseases, but we have tried here to take advantage of this complexity by looking at gene interactions and signaling pathways between MS and OCD, due to their important clinical consequence. Here, thirty-two shared genes were detected between MS and OCD disorders. The highest degree and maximum betweenness centrality as the topological features have been shown in different categories of genes, such as neurotrophic factor (e.g., BDNF), inflammatory cytokines (i.e., IL-6 and TNF), apoptotic factor (e.g., caspase-3), and cellular responses (e.g., MAPK1 and ESR1). In addition to the current interactions, we also identified ten novel genes with the STRING database that have significant interactions with 32 shared genes between MS and OCD. Some of the ten novel genes have not been previously studied in the context of MS-OCD; therefore, they may play a role in comorbidity interactions and may have important pathogenic mechanisms for MS-OCD. As a novel part of our study, five transcription factors and twenty-five miRNAs were predicted related to ten genes that had more connection with common genes. Among ten genes, *STAT3* and *NTRK2* had the highest connections with the shared genes. They had the greatest centrality with novel genes. Moreover, these two genes had the highest connection from enrichment results (i.e., molecular function and pathways). Furthermore, main signaling pathways, such as immune interaction, cytokine responses, and disruption of receptor signaling pathways have been predicted for *STAT3* and *NTRK2* in the context of MS-OCD.

MS and OCD are highly genetic complaints that are assumed to share inherent risk factors. A review article by Enders et al. proposed the ”autoimmune-OCD subtype” as it has been known that a subgroup of patients may have a secondary form of OCD with an organic cause and interestingly, autoimmune disorders are frequently associated with the secondary form of OCD [27]. To date, the identification of decisive vulnerability genes for these etiologically multifaceted disorders remains indefinable. Here, we reported the first comprehensive bioinformatics analysis of the relationship between MS and OCD. Finding common genes between MS and OCD is important to figure out mechanisms and downstream signaling for novel therapeutic options, but this approach is not enough. Focusing on genes that have more connections with shared genes is needed. To this point, we also found out two genes (i.e., *STAT3* and *NTRK2*) that have the highest topological features with common genes between MS and OCD.

Our bioinformatics analysis also predicted the neurotrophic tyrosine kinase receptor type 2 (*NTRK2*) gene that had more connections with common genes. *NTRK2* encodes for the protein tropomyosin receptor kinase B (TrkB), which is a neurotrophin receptor with a high affinity for BDNF and contributes to several physiological functions of neurons, including cell survival and differentiation [28]. Genetic susceptibility of the BDNF/NTRK2 signaling pathway was reported in OCD [29], but this pathway has not been investigated in the context of MS. To have a wide view of the function of *NTRK2*, miR-339-5p and miR-2116-3p were predicted. It has been reported that miR-339-5p modulated the expression of pro-inflammatory markers (i.e., IL-1β, IL-6, and TNF-α) through the inhibition of the NF-κB pathway [30]. Therefore, it is suggested that predicted miRNAs can be targeted for therapeutic options and investigated as biomarkers in MS-OCD in connection with the stability of miRNAs in body fluids. Therefore, the present investigation is part of ongoing research to explain the genetic components involved in the etiology of MS and OCD characteristics. 

Another predicted gene is *STAT3* which has 19 connections with common genes. Recently, it has been shown that STAT3 signaling in myeloid cells stimulates pathogenic myelin-specific T cell differentiation and autoimmune demyelination [31]. The role of STAT3 in mood disorders has been indicated by several lines of evidence in terms of STAT3 activity, serotonergic neurotransmission, and the control of behaviors relevant to psychopathology [32]; however, evidence for STAT3 in the course of OCD is still limited. A recent bioinformatic study by de Oliveira et al. predicted STAT3 as a significant transcription factor in relation to OCD [33]. We also predicted four miRNAs, such as miR-21-5p, miR-32-3p, miR-347a-3p, and miR-590-5p, for the *STAT3* gene. Regarding our data, identifying the role of the *STAT3* gene and its epigenetic modifications in MS patients’ coexistence with OCD is suggested for future studies. Our model also predicted two anti-inflammatory cytokines with the highest connections. Based on the literature review, levels of IL-4 and IL-10 have not changed in OCD patients [34], while these cytokines have been involved in the immunopathogenesis of MS [35]. Further investigation and experimentation into anti-inflammatory cytokines in MS-OCD are strongly recommended.

To translate gene interactions into signaling pathways and molecular function, we further performed enrichment analyses on two top genes (i.e., *STAT3* and *NTRK2*). Our data showed that the main disrupted signaling pathways were immune interaction, cytokine responses, and disruption of receptor signaling pathway *STAT3* and *NTRK2* in the context of MS-OCD. We also demonstrated that protein homodimerization activity was predicted for *STAT3* and *NTRK2*. Furthermore, protein kinase binding and primary miRNA binding were predicted for *STAT3*, while transmembrane receptor protein kinase activity and neurotrophin binding were predicted for *NTRK2* in terms of molecular function. The molecular function and signaling pathway related to our target genes can provide us valuable data for novel mechanisms. Therefore, this bioinformatics study provides a good starting point for further research at experimental and clinical grades. However, it should be noted that this investigation is a bioinformatics study; if the debate is to be moved forward, a better understanding can be achieved by experimental research on this topic.

## 5. Conclusions

On this basis, we conclude that the co-occurrence of MS and OCD is related to genetic interactions; therefore, we performed different levels of analysis to predict the main gene’s connection and their epigenetic modifications. Interestingly, our bioinformatics results indicated that some genes have not been investigated in MS and OCD experimentally and clinically yet. We introduced *STAT3* and *NTRK2* genes that had the highest connections and performed further enrichment analysis for their signaling pathways and molecular functions. Future studies into the shared genetic relations between MS and OCD will present opportunities for researchers to build an agenda to address the challenges of disorder etiologies. Finally, postmortem and clinical studies (i.e., cohort and retrospective studies) can be performed to indicate the role of predicted genes. In postmortem studies, we can detect the expression of novel genes in the brain areas that are involved in MS and OCD. In cohort or case–control studies, we can assess the expression of miRNAs related to *STAT3* and *NTRK2* genes in serum or CSF samples as novel biomarkers. 

## Figures and Tables

**Figure 1 jpm-12-01043-f001:**
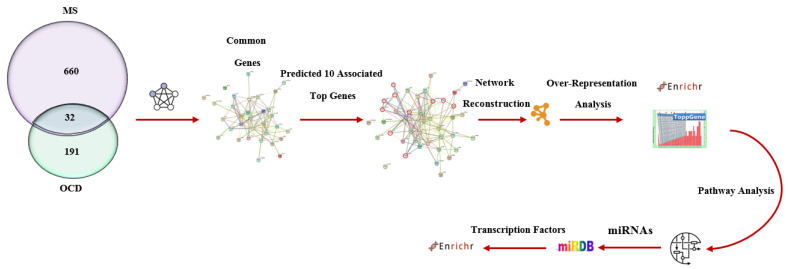
Flowchart of the main steps and bioinformatic tools in the current study. All genes associated with multiple sclerosis (MS) and obsessive-compulsive disorder (OCD) were extracted from the literature review and Harmonizome database. Then, 32 shared genes were identified between the two diseases. Next, novel genes based on protein–protein interactions with the highest connections with shared gene sets were predicted by the STRING database. The obtained genetic network was uploaded into Cytoscape to reconstruct a co-expression novel genetic network in a background of shared genes. Network parameters were also calculated through the Network Analyzer Toolkit in Cytoscape. All enrichment analysis was conducted on 10 predicted novel predicted genes.

**Figure 2 jpm-12-01043-f002:**
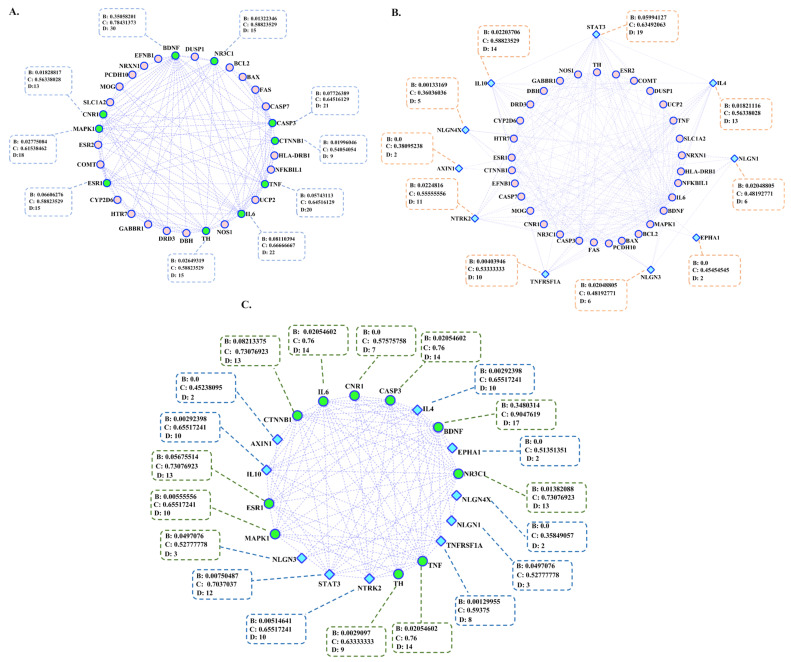
Reconstructed genetic networks for MS and OCD. (**A**) Genetic network for MS-OCD shared genes. The current network is comprised of 32 nodes (genes) and 121 edges (interactions). Green nodes represent genes with greater network features (hub genes in the network; network parameters: density = 0.254, diameter = 5, centralization = 0.513, clustering coefficient = 0.583). (**B**) Genetic network for common and ten top genes between MS and OCD. Circular pink nodes represent shared genes and external blue diamond nodes represent novel predicted genes for MS and OCD (network parameters: density = 0.239, diameter = 5, centralization = 0.537, clustering coefficient = 0.662). (**C**) Genetic networks for 10 genes with higher topological features between MS and OCD shared genes (green nodes) and 10 novel predicted genes (blue nodes) (network parameters: density = 0.489, diameter = 4, centralization = 0.450, clustering coefficient = 0.836). B: betweenness centrality; C: closeness centrality; D: degree.

**Figure 3 jpm-12-01043-f003:**
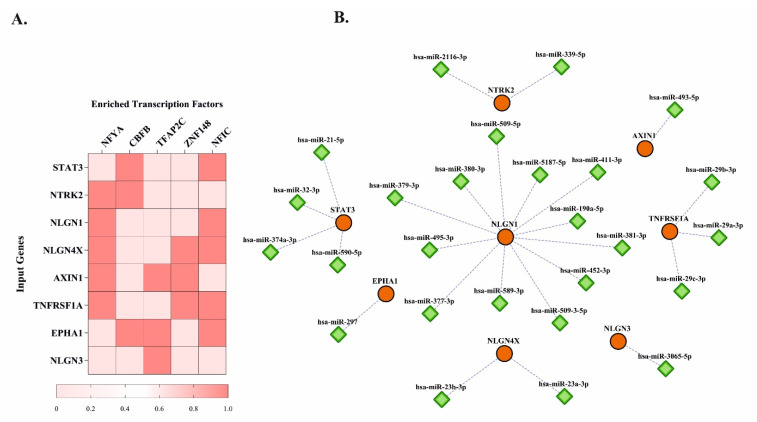
Heat map of predicted transcription factors and reconstructed gene-miRNAs network for predicted novel genes associated with both MS and OCD. (**A**) Each gene with its enriched transcription factor is adjusted with a color map. Each column represents an enriched transcription factor for some genes in the rows. Red squares specify positive enrichment (1 score), and pink squares indicate negative enrichment (0 score). Each enriched transcription factor is ranked according to the highest order of importance (*p*-value) from left to right. (**B**) Each gene (circle nodes) is connected to its enriched miRNAs (diamond nodes) through dotted lines. All enriched miRNAs have a target prediction score of greater than 90 percent according to the miRDB database.

**Figure 4 jpm-12-01043-f004:**
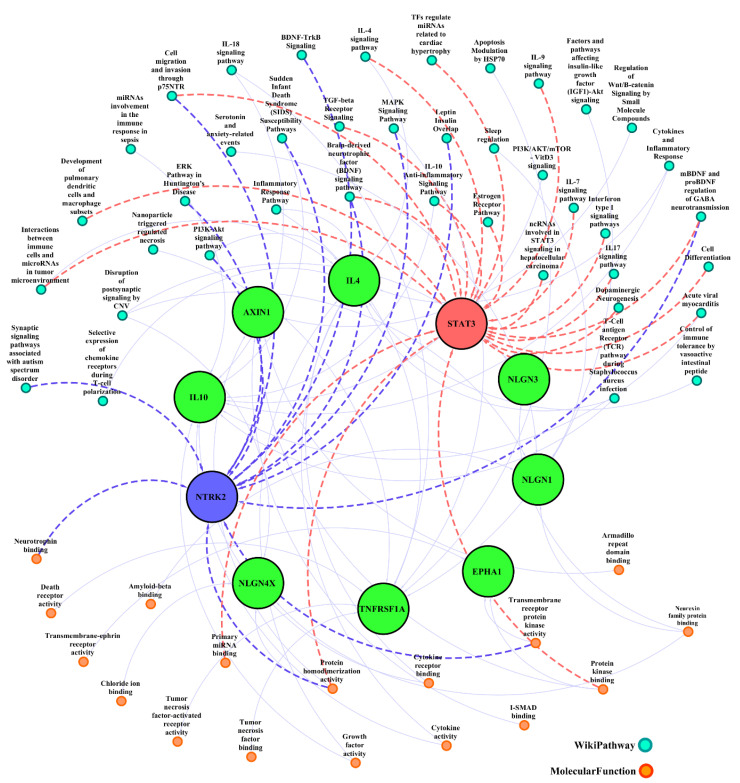
Molecular and pathway genetic network of 10 predicted novel genes for MS and OCD. Central big nodes indicated 10 predicted novel genes for MS and OCD. Peripheral orange nodes represented all enriched molecular functions and turquoise nodes specified main involved pathways for 10 central novel genes. *STAT3* and *NTRK2* are two important genes with the highest degree (more connected) in the network and annotated most of the enrichments. All enriched categories are connected to their target genes and are annotated as enrichment *p*-value less than 0.05.

**Table 1 jpm-12-01043-t001:** The number of obtained genes from different search strategies.

Number of Genes Associated with Diseases		
Gene Source	Multiple Sclerosis	Obsessive–Compulsive Disorder
Literature review	368	133
Harmonizome	292	58
Total genes	660	191
Shared genes	32	
Hub genes (Acquired by STRING)	10	

**Table 2 jpm-12-01043-t002:** Biological process enrichment results for predicted novel genes of multiple sclerosis (MS) and obsessive-compulsive disorder (OCD).

Description	Target Genes	−log (*p*-Value)
Presynaptic membrane assembly	NLGN3, NLGN1, NLGN4X	7.871923987
Postsynaptic membrane assembly	NLGN3, NLGN1, NLGN4X	7.747146969
Presynaptic membrane organization	NLGN3, NLGN1, NLGN4X	7.528708289
Regulation of chronic inflammatory response	IL10, TNFRSF1A, IL4	7.178551981
Neuron cell–cell adhesion	NLGN3, NLGN1, NLGN4X	7.178551981
Positive regulation of developmental process	NLGN3, NLGN1, NTRK2, STAT3, IL4, IL10, EPHA1, TNFRSF1A	7.108016769
Regulation of multicellular organismal development	NLGN3, NLGN1, NTRK2, STAT3, IL4, IL10, EPHA1, TNFRSF1A	6.813326133
Positive regulation of signal transduction	NLGN3, NLGN1, NTRK2, STAT3, IL4, IL10, AXIN1, TNFRSF1A	6.771086594
Positive regulation of multicellular organismal process	NLGN3, NLGN1, NTRK2, STAT3, IL4, IL10, EPHA1, TNFRSF1A	6.753009301
Regulation of anatomical structure morphogenesis	NLGN3, NLGN1, NTRK2, STAT3, IL10, EPHA1, TNFRSF1A	6.378512135
Positive regulation of synaptic transmission, glutamatergic	NLGN3, NLGN1, NTRK2	6.16627931
Postsynapse assembly	NLGN3, NLGN1, NLGN4X	6.16627931
Chronic inflammatory response	IL10, TNFRSF1A, IL4	6.1316495
Peptidyl-tyrosine phosphorylation	NTRK2, STAT3, IL4, EPHA1, TNFRSF1A	6.070274622
Peptidyl-tyrosine modification	NTRK2, STAT3, IL4, EPHA1, TNFRSF1A	6.05527064
Cell junction organization	NLGN3, NLGN1, NLGN4X, IL10, EPHA1, NTRK2	5.919734373
Behavior	NLGN3, NLGN1, NTRK2, STAT3, AXIN1, NLGN4X	5.806041022
Postsynaptic membrane organization	NLGN3, NLGN1, NLGN4X	5.775208044
Receptor signaling pathway via JAK-STAT	IL10, TNFRSF1A, IL4, STAT3	5.767766479
Regulation of nervous system process	NLGN3, NLGN1, NLGN4X, IL10	5.72514968
Receptor signaling pathway via STAT	IL10, TNFRSF1A, IL4, STAT3	5.699839463
Cell junction assembly	NLGN3, NLGN1, NLGN4X, NTRK2, EPHA1	5.661145254
Regulation of tumor necrosis factor production	IL10, TNFRSF1A, IL4, STAT3	5.634699251
Negative regulation of reactive oxygen species biosynthetic process	IL10, IL4, STAT3	5.628193541
Blood vessel morphogenesis	IL10, TNFRSF1A, EPHA1, STAT3, NTRK2, NLGN1	5.626168855
Regulation of tumor necrosis factor superfamily cytokine production	IL10, TNFRSF1A, IL4, STAT3	5.595166283
Tumor necrosis factor production	IL10, TNFRSF1A, IL4, STAT3	5.587371479
Modulation of excitatory postsynaptic potential	NLGN3, NLGN1, NLGN4X	5.56050941
Presynapse assembly	NLGN3, NLGN1, NLGN4X	5.56050941
Positive regulation of angiogenesis	IL10, TNFRSF1A, EPHA1, STAT3	5.548981548
Positive regulation of vasculature development	IL10, TNFRSF1A, EPHA1, STAT3	5.548981548
Tumor necrosis factor superfamily cytokine production	IL10, TNFRSF1A, IL4, STAT3	5.541362151
Negative regulation of chronic inflammatory response	IL10, IL4	5.500725418
Positive regulation of peptidyl-tyrosine phosphorylation	IL4, TNFRSF1A, STAT3, NTRK2	5.496481687
Regulation of cell junction assembly	NLGN3, NLGN1, NTRK2, EPHA1	5.481881053
Synapse assembly	NLGN3, NLGN1, NLGN4X, NTRK2	5.481881053
Synapse organization	NLGN3, NLGN1, NLGN4X, NTRK2, IL10	5.430977414
Positive regulation of phosphorylation	IL4, NTRK2, STAT3, AXIN1, EPHA1, TNFRSF1A	5.377785977

## Data Availability

The datasets generated and/or analyzed during the current study are available from the corresponding authors upon reasonable request.

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
