# Peer review of "STAT3 and NTRK2 Genes Predicted by the Bioinformatics Approach May Play Important Roles in the Pathogenesis of Multiple Sclerosis and Obsessive–Compulsive Disorder"

_jpm, 2022, doi:10.3390/jpm12071043_

Round 1
Reviewer 1 Report
Dear Author,
Thanks for submitting your research manuscript entitled "Novel genes predicted by the bioinformatics approach may 2 play important roles in the pathogenesis of multiple sclerosis 3 and obsessive-compulsive disorder ".
Before giving my final comments as well as the final revision of this manuscript, the author needs to address the following comments in a scientific manner.
Major concerns:-
Please find out the following comments
Title, Abstract, Introduction:
- Title need to be very specific. is not acceptable in current form.
- The rationale and purpose behind the correlation between OCD and MS is not clear as well as incomplete.
- Author must start their introduction direct about the correlation and rationale behind this study instead writing about the common information regarding MS and OCD.
- Abstract is very confusing. Irrational and fused with repetitions. Scientific output is not clear with this abstract.
- The reviewer feels author need to elaborate and justify it with proper citations with strong evidences. Author fails to explain the relevant justification in the introduction as well as mentioned in the discussion part.
- A major drawback is a lack of clinical evidence, and incomplete experimental design is the main concern.
- Reviewer found irrational and non-scientific justification in abstract. Introduction, as well as in discussion part.
- Authors fail to justify the correlation of cellular and molecular target BDNF, STAT3, and NTRK2 and their associated genes
- Selection of target gene in material and method section is incomplete.
Results:
- Results need more clarification and significant justification. Differentiating between the outcome and the discussion sections is quite difficult.
- It looks like; there is no scientific clarity, and follow continuous paragraph writing without any point
- Complete mismatch of figures. No visible, poorly explained. Need major revision.
Discussion:
- To address the outcome of measures/results separately and how they correlate with the existing literature, it would be better if the author restructured to take a more critical approach.
- In both the discussion and the conclusion, the aims, rationale, and future perspectives are not evident clearly in relation with in-vitro experimentation.
- The discussion is usually organized at the beginning to address all the observations and evaluate them at the end. It makes the results easier to contextualize and simpler to comprehend. Furthermore, a minimal critical analysis should be provided
- Add limitations of this study at the end of the discussion part.
Conclusion:
- Need to revise the conclusion in a scientific manner. No accepted in current form.
- This reviewer considers that this paper cannot be published in the present form. A detailed revision shortening, ordering and following the commented ideas could improve this interesting paper in a significant manner.
- Several typewriting mistakes are present and needing correction. This reviewer remains at entire disposal for the next version.
Author Response
14th, June 2022
To: Journal of Personalized Medicine, Editors
We wish to express our appreciation to the reviewers for their insightful comments, which have helped us significantly to improve our manuscript. We have revised our paper accordingly and feel that your comments helped clarify and improve our paper. Please find our response (in blue) to the reviewer’s specific comments (in black) below. In the submitted revised manuscript, we have highlighted the revised text in green color.
Reviewer 1
Title, Abstract, Introduction:
Title need to be very specific. is not acceptable in current form.
Our response: As your suggestion, the title was revised to present the content of the text.
The rationale and purpose behind the correlation between OCD and MS is not clear as well as incomplete. Author must start their introduction direct about the correlation and rationale behind this study instead writing about the common information regarding MS and OCD.
Our response: Although our knowledge of human diseases has increased dramatically, the molecular basis, phenotypic traits, and therapeutic targets of most diseases still remain unclear. An increasing number of studies have observed that comorbidities in diseases often are caused by similar molecules, can be diagnosed by similar markers or phenotypes, or can be cured by similar drugs. Thus, the identification of diseases similar to known ones has attracted considerable attention worldwide. To this end, the associations between diseases at the molecular and genetic were used to measure the pairwise similarity in diseases.
In this study, we wanted to indicate the common genetic underprints between MS and OCD because there are no data available at this level. We cited clinical studies that have shown there are correlations between two diseases but the molecular underpinnings are not clear. Therefore, the genetic contribution to OCD and MS can get us valuable information for the next therapeutic options, e.g., personalized medicine. Thus, the purpose of this study was to find out novel genes with the highest centrality by bioinformatics approach because these genes may be involved in the pathogenesis. This study can be used as an initiator for experimental research on genetic manipulation or biomarkers. Finally, to make the manuscript highly readable as you suggested, we revised the abstract, introduction, and discussion.
Abstract is very confusing. Irrational and fused with repetitions. Scientific output is not clear with this abstract.
Our response: As your suggestion, the abstract was revised. For the sake of word limitations, we put the essential data in the abstract. The rationale of this study was also added to the first part of the abstract.
The reviewer feels author need to elaborate and justify it with proper citations with strong evidences. Author fails to explain the relevant justification in the introduction as well as mentioned in the discussion part.
Our response: Several similar studies have been cited in the text.
A major drawback is a lack of clinical evidence, and incomplete experimental design is the main concern.
Our response: Actually, there are very few studies on this topic at the level of the common genetic network; however, the current experimental and clinical studies were cited in the introduction. The novelty part is that there is a correlation between MS and OCD from retrospective and prospective data, but the underlying mechanisms are not clear.
For the second part of the comment, I should mention the current study is a bioinformatics investigation, we didn‘t have any experimental design.
Reviewer found irrational and non-scientific justification in abstract. Introduction, as well as in discussion part.
Our response: I’m not so sure about that. From a structural and design view, the current study is in accord with similar studies (Kawsar et al. 2020; Kang et al. 2022; Sepehrinezhad et al. 2021). For the purpose and rationale of the study, we revised the introduction and discussion.
Md Kawsar, Tasnimul Alam Taz, Bikash Kumar Paul, Shahin Mahmud, Md Manowarul Islam, Touhid Bhuyian, Kawsar Ahmed, Analysis of gene network model of Thyroid Disorder and associated diseases: A bioinformatics approach, Informatics in Medicine Unlocked, Volume 20, 2020, 100381, ISSN 2352-9148, https://doi.org/10.1016/j.imu.2020.100381.
Kang J, Kwon EJ, Ha M, Lee H, Yu Y, Kang JW, Kim Y, Lee EY, Joo JY, Heo HJ, Kim EK, Kim TW, Kim YH, Park HR. Identification of Shared Genes and Pathways in Periodontitis and Type 2 Diabetes by Bioinformatics Analysis. Front Endocrinol (Lausanne). 2022 Jan 25;12:724278. doi: 10.3389/fendo.2021.724278. PMID: 35145474; PMCID: PMC8822582.
Sepehrinezhad A, Rezaeitalab F, Shahbazi A, Sahab-Negah S. A Computational-Based Drug Repurposing Method Targeting SARS-CoV-2 and its Neurological Manifestations Genes and Signaling Pathways. Bioinform Biol Insights. 2021 Jun 18;15:11779322211026728. doi: 10.1177/11779322211026728. PMID: 34211268; PMCID: PMC8216348.
Authors fail to justify the correlation of cellular and molecular target BDNF, STAT3, and NTRK2 and their associated genes.
Our response: Thank you for your advice. Since STAT3 and NTRK2 had more connections in terms of molecular function and pathways, we drew a novel concept map, as shown in Fig 4.
Selection of target gene in material and method section is incomplete.
Our response: We revised this part of the manuscript.
Results:
Results need more clarification and significant justification. Differentiating between the outcome and the discussion sections is quite difficult. It looks like; there is no scientific clarity, and follow continuous paragraph writing without any point. Complete mismatch of figures. No visible, poorly explained. Need major revision.
Our response: thank you for your suggestion. We redesigned the results. In our study, We had two genetic networks. The first network is the common genes between MS and OCD, and the second network is the genes that have the highest connections with shared genes. We predicted transcription factors, miRNAs, and biological functions for the second network. The revised part is that we predicted molecular functions and pathways for novel genes individually. As shown in Fig 4, STAT3 and NTRK2 had more connections in terms of molecular functions and pathways; therefore, we just focus on these genes in the rest of the article.
I see what you’re saying but the current investigation is a bioinformatics study. We should point out the major data in the result part. Regarding the flowchart of the main steps and bioinformatic tools, as shown in figure 1, the results have been designed.
Discussion:
To address the outcome of measures/results separately and how they correlate with the existing literature, it would be better if the author restructured to take a more critical approach.
In both the discussion and the conclusion, the aims, rationale, and future perspectives are not evident clearly in relation with in-vitro experimentation.
Our response: This study is a bioinformatics investigation, so this study is an initiative. In the current study, we used existing data from different levels of analysis to show there are correlations between MS and OCD. In our study, we just predicted some genes, epigenetic, and molecular underpinnings from big data that may play in the pathogenesis of the two diseases. We don‘t want to compare our results with in vitro studies. Furthermore, we couldn‘t find any study at the in vitro level that show common mechanisms between MS and OCD.
The discussion is usually organized at the beginning to address all the observations and evaluate them at the end. It makes the results easier to contextualize and simpler to comprehend. Furthermore, a minimal critical analysis should be provided
Our response: The structure of our study was revised in an academic manner. The main findings of our study are presented in the first paragraph of the discussion. Then, in the main text of the discussion, relevant studies were discussed with our results.
Add limitations of this study at the end of the discussion part.
Our response: It was added at the end of the discussion.
Conclusion:
Need to revise the conclusion in a scientific manner. No accepted in current form.
Our response: The conclusion has been written in an academic manner. We started with the restatement of our study, then we summarized our findings, and finally, we suggested future work. If you have any alternative way, I would be very happy to hear you.
Our conclusion was divided into the following parts:
Restating the aims of the study
Summarising the main research findings
Making recommendations for further research work
On this basis, we conclude that the co-occurrence of MS and OCD is related to genetic interactions; therefore, we performed different levels of analysis to predict the main gene’s connection and their epigenetic modifications. Interestingly, our bioinformatics results indicated that some genes have not been investigated in MS and OCD experimentally and clinically yet. We introduced STAT3 and NTRK2 genes that had the highest connections and performed further enrichment analysis for their signaling pathways and molecular functions. Future studies into the shared genetic relations between MS and OCD will present opportunities for researchers to build an agenda to address the challenges of disorder etiologies. Finally, postmortem and clinical studies (i.e., cohort and retrospective studies) can be done to indicate the role of predicted genes. In postmortem studies, we can detect the expression of predicted genes in the brain areas that are involved in MS and OCD. In cohort or case-control studies, we can assess the expression of miRNAs related to STAT3 and NTRK2 genes in serum or CSF samples as novel biomarkers.
This reviewer considers that this paper cannot be published in the present form. A detailed revision shortening, ordering and following the commented ideas could improve this interesting paper in a significant manner. Several typewriting mistakes are present and needing correction. This reviewer remains at entire disposal for the next version.
Our response: As your suggestions, we revised critically the manuscript. We just focused on genes that have the highest centrality (Fig 4). Furthermore, the introduction, results, and discussion were revised.
Sincerely,
Dr. Sajad Sahab Negah,
Department of Neuroscience, Mashhad University of Medical Sciences, Mashhad, Iran.
Email: sahabnegahs@mums.ac.ir;Tel: +98-51-38002473

Reviewer 2 Report
I consider that your work is a work with some interesting strengths. I believe that currently the search for the mechanisms of action involved in the pathophysiology of many mental illnesses is the focus of research. I also consider that bioinformatic processes represent the first steps for the approximation of theories and searches for molecular mechanisms.
From my point of view, visual points of your article should be improved, for example Figure 2. Reconstructed genetic networks for MS and OCD. I can't see the letters in the image and reading is difficult.
The brief definition of certain processes to bring reading closer to a greater number of potential readers may also be interesting. If the idea is to focus on people specialized in the subject, I understand that it would not be necessary.
Author Response
14th, June 2022
To: Journal of Personalized Medicine, Editors
We wish to express our appreciation to the reviewers for their insightful comments, which have helped us significantly to improve our manuscript. We have revised our paper accordingly and feel that your comments helped clarify and improve our paper. Please find our response (in blue) to the reviewer’s specific comments (in black) in the uploaded PDF file. In the submitted revised manuscript, we have highlighted the revised text in green color.
Reviewer 2
I consider that your work is a work with some interesting strengths. I believe that currently the search for the mechanisms of action involved in the pathophysiology of many mental illnesses is the focus of research. I also consider that bioinformatic processes represent the first steps for the approximation of theories and searches for molecular mechanisms.
From my point of view, the visual points of your article should be improved, for example Figure 2. Reconstructed genetic networks for MS and OCD. I can't see the letters in the image and reading is difficult.
Our response: We considered this issue and improved the quality of figures in the revised version.
The brief definition of certain processes to bring reading closer to a greater number of potential readers may also be interesting. If the idea is to focus on people specialized in the subject, I understand that it would not be necessary.
Our response: Thanks for your consideration. All used methods and procedures are approved according to previous bioinformatic studies and can express our goals.
Sincerely,
Dr. Sajad Sahab Negah,
Department of Neuroscience, Mashhad University of Medical Sciences, Mashhad, Iran.
Email: sahabnegahs@mums.ac.ir;Tel: +98-51-38002473

Reviewer 3 Report
This review suggests that bioinformatics approach may play important roles in the pathogenesis of multiple sclerosis and obsessive-compulsive disorder. I suggest some revision for improving quality of study.
(Comment 1) I recommend authors to supplement the full name for the first abbreviation used in the abstract (e.g. OCD) and manuscript (e.g. MAOA).
(Comment 2) I recommend authors to change reference 4 to the latest research. The presented variations of prevalence (15~30%) is too large. (line 51-52)
(Comment 3) Authors mentioned "Bioinformatics methods on the molecular genetic association can yield greater power". I recommend authors to supplement this statement with more detalis in the introduction section. (line 69)
Dicussion
(Comment 4) This study analyzed the relationship between MS and OCD in bioinformatics. In order not to stop at the interpretation of the results of a simple analysis, what kind of research should be conducted in the future based on the results of this analysis, the specific direction and method should be suggested.
Author Response
14th, June 2022
To: Journal of Personalized Medicine, Editors
We wish to express our appreciation to the reviewers for their insightful comments, which have helped us significantly to improve our manuscript. We have revised our paper accordingly and feel that your comments helped clarify and improve our paper. Please find our response (in blue) to the reviewer’s specific comments (in black) in the uploaded PDF file. In the submitted revised manuscript, we have highlighted the revised text in green color.
Reviewer 3
This review suggests that bioinformatics approach may play important roles in the pathogenesis of multiple sclerosis and obsessive-compulsive disorder. I suggest some revision for improving quality of study.
(Comment 1) I recommend authors to supplement the full name for the first abbreviation used in the abstract (e.g. OCD) and manuscript (e.g. MAOA).
Our response: We considered your valuable comment and revised the full name of all abbreviations in the revised version.
(Comment 2) I recommend authors to change reference 4 to the latest research. The presented variations of prevalence (15~30%) is too large. (line 51-52)
Our response: We updated the newest data on the incidence of OCD in MS in the revised manuscript.
(Comment 3) Authors mentioned "Bioinformatics methods on the molecular genetic association can yield greater power". I recommend authors to supplement this statement with more detalis in the introduction section. (line 69)
Our response: This statement was revised and highlighted in the context.
Dicussion
(Comment 4) This study analyzed the relationship between MS and OCD in bioinformatics. In order not to stop at the interpretation of the results of a simple analysis, what kind of research should be conducted in the future based on the results of this analysis, the specific direction and method should be suggested.
Our response: We believe that further validation through experiments on these genes is necessary. Two types of studies can be designed. Postmortem investigations and clinical studies (i.e., cohort and retrospective studies) can be done to indicate the role of predicted genes. In the postmortem studies, we can detect the expression of genes in the different brain areas that we predicted. In cohort or case-control studies, we can assess the expression of miRNA in serum or CSF samples. As your effective suggestion, the above-mentioned sentences were added to the conclusion part.
Sincerely,
Dr. Sajad Sahab Negah,
Department of Neuroscience, Mashhad University of Medical Sciences, Mashhad, Iran.
Email: sahabnegahs@mums.ac.ir;Tel: +98-51-38002473

Round 2
Reviewer 1 Report
Dear author,
After careful revision, reviewer feel that manuscript can be accepted for publication.